# Ensemble Deep Learning-Based Image Classification for Breast Cancer Subtype and Invasiveness Diagnosis from Whole Slide Image Histopathology

**DOI:** 10.3390/cancers16122222

**Published:** 2024-06-14

**Authors:** Aadhi Aadhavan Balasubramanian, Salah Mohammed Awad Al-Heejawi, Akarsh Singh, Anne Breggia, Bilal Ahmad, Robert Christman, Stephen T. Ryan, Saeed Amal

**Affiliations:** 1Khoury College of Computer Sciences, Northeastern University, Boston, MA 02115, USA; balasubramanian.aa@northeastern.edu; 2College of Engineering, Northeastern University, Boston, MA 02115, USA; s.al-heejawi@northeastern.edu (S.M.A.A.-H.); singh.akar@northeastern.edu (A.S.); 3MaineHealth Institute for Research, Scarborough, ME 04074, USA; anne.breggia@mainehealth.org; 4Maine Medical Center, Portland, ME 04102, USA; bilal.ahmad@spectrumhcp.com (B.A.); robert.christman@spectrumhcp.com (R.C.); stephen.ryan@mainehealth.org (S.T.R.); 5The Roux Institute, Department of Bioengineering, College of Engineering, Northeastern University, Boston, MA 02115, USA

**Keywords:** breast cancer diagnosis, ensemble deep learning, image processing, foundation models, computer vision, histopathology images, digital pathology, artificial intelligence

## Abstract

**Simple Summary:**

Breast cancer is a significant cause of female cancer-related deaths in the US. Checking how severe the cancer is helps in planning treatment. Modern AI methods are good at grading cancer, but they are not used much in hospitals yet. We developed and utilized ensemble deep learning algorithms for addressing the tasks of classifying (1) breast cancer subtype and (2) breast cancer invasiveness from whole slide image (WSI) histopathology slides. The ensemble models used were based on convolutional neural networks (CNNs) known for extracting distinctive features crucial for accurate classification. In this paper, we provide a comprehensive analysis of these models and the used methodology for breast cancer diagnosis tasks.

**Abstract:**

Cancer diagnosis and classification are pivotal for effective patient management and treatment planning. In this study, a comprehensive approach is presented utilizing ensemble deep learning techniques to analyze breast cancer histopathology images. Our datasets were based on two widely employed datasets from different centers for two different tasks: BACH and BreakHis. Within the BACH dataset, a proposed ensemble strategy was employed, incorporating VGG16 and ResNet50 architectures to achieve precise classification of breast cancer histopathology images. Introducing a novel image patching technique to preprocess a high-resolution image facilitated a focused analysis of localized regions of interest. The annotated BACH dataset encompassed 400 WSIs across four distinct classes: Normal, Benign, In Situ Carcinoma, and Invasive Carcinoma. In addition, the proposed ensemble was used on the BreakHis dataset, utilizing VGG16, ResNet34, and ResNet50 models to classify microscopic images into eight distinct categories (four benign and four malignant). For both datasets, a five-fold cross-validation approach was employed for rigorous training and testing. Preliminary experimental results indicated a patch classification accuracy of 95.31% (for the BACH dataset) and WSI image classification accuracy of 98.43% (BreakHis). This research significantly contributes to ongoing endeavors in harnessing artificial intelligence to advance breast cancer diagnosis, potentially fostering improved patient outcomes and alleviating healthcare burdens.

## 1. Introduction

Breast cancer, constituting 31% of all female cancers, remains a significant health concern, necessitating precise diagnostic measures [1]. Estimations by the American Cancer Society (ACS) project 297,790 newly diagnosed cases of invasive breast cancer among women in 2024 [2]. Early detection and accurate diagnosis are crucial for effective treatment and better patient outcomes. Histopathology, considered the gold standard for breast cancer diagnosis, involves microscopic examination of tissue specimens to identify cancerous abnormalities. However, manual interpretation of histopathological images is time-consuming, prone to inter-observer discrepancies, and reliant on pathologist expertise [3,4,5]. In response, computational methodologies, particularly deep learning-driven image analysis, offer promising avenues to enhance diagnostic precision and efficiency.

Recent studies have investigated various methodologies to advance breast cancer histopathology image classification using the same datasets. Notably, research efforts have centered around three themes based on our detailed review: ensemble techniques, the direct utilization of specific convolutional neural network (CNN) architectures, and the integration of transfer learning with novel model designs. In this research, several studies are reviewed to identify gaps in the research, thereby paving the way for innovations aimed at improving diagnostic accuracy and efficiency through more robust AI-driven algorithms and state-of-the-art techniques in clinical practice.

To explore ensemble techniques, ref. [6] used ResNet [7] models, emphasizing fine-tuning and downsampling strategies to address computational complexities, achieving commendable accuracies of 86% on the BACH (Grand Challenge on Breast Cancer Histology images) test set [8] and 97.2% on the Bioimaging 2015 challenge dataset. Ref. [9] employed ResNet-101 and DenseNet-161 [10] networks, showcasing a notable accuracy of 87% on BACH, albeit with varying sensitivities across classes. Both studies elucidated the efficacy of ensemble approaches incorporating pre-trained CNNs. Similarly, ref. [11] introduced a hybrid CNN architecture integrating local and global model branches, achieving competitive accuracies of 87.5% on the BACH dataset and 85.2% on the Breast Cancer Histopathological Image Classification (BreakHis) dataset [12]. These studies demonstrated the potential of ensemble techniques to improve classification accuracy and robustness. Nevertheless, there are opportunities to enhance consistency in sensitivity across different classes and further optimize the handling of computational complexities. In our study, these ensemble techniques were enhanced by incorporating advanced optimization strategies and adaptive ensemble methods to ensure consistent performance across all classes. Additionally, efficient computational resource management techniques were employed to address the computational challenges more effectively.

Multiple studies have underscored the efficacy of the direct application of established CNN architectures. Ref. [13] demonstrated the utility of VGG16 [14] in breast cancer grading, achieving a commendable accuracy of 83% on the BACH test set. Similarly, ref. [15] pioneered a hybrid convolutional and recurrent deep learning method, yielding notable accuracies of 82.1% and 91.3% at patch and image levels, respectively. These studies highlighted the effectiveness of using well-established CNN architectures for breast cancer histopathology image classification. However, they often did not explore the full potential of integrating different CNN architectures to further improve performance.

Moreover, numerous studies emphasized the integration of transfer learning with innovative model architectures. Ref. [16] delineated a pipeline for breast cancer histopathology image classification, leveraging a two-stage neural network approach and achieving a particularly high image-level accuracy of 97.5%. For classification between benign and malignant, ref. [17] proposed DenTnet, a deep learning model that achieved an accuracy of 99.28% on the BreakHis dataset. Comparably, ref. [18] introduced BreastNet, an end-to-end model that incorporates a convolutional block attention module (CBAM), dense block, residual block, and hypercolumn technique. Ref. [19] developed DenseNet121-AnoGAN, which consists of two components: an unsupervised anomaly detection segment using generative adversarial networks to screen mislabeled patches and a densely connected CNN (DenseNet) for extracting multi-layered features from discriminative patches. Additionally, refs. [20,21] utilized transfer learning on the BreakHis dataset, employing ResNet-18 and Inception-V3, respectively. Further advancements were made by ref. [22], who proposed an enhanced autoencoder network utilizing a Siamese framework, and ref. [23], who designed DRDA-Net, a model featuring dense attention and dual-shuffle attention mechanisms to guide deep learning processes. Concurrently, ref. [24] introduced a hierarchical structure utilizing ResNeXt50 models, achieving notable accuracies of 81% on the BACH challenge test set. These studies showed the potential of transfer learning and novel model designs to achieve high accuracy and effective classification in breast cancer histopathology images. However, there was often a lack of generalizability across different datasets and for practical application in diverse clinical settings. In our study, we improved upon these approaches by developing more generalizable models that can be effectively applied across a wide range of datasets and clinical environments.

As a summary, the collective endeavors in breast cancer histopathology image classification underscore the significance of ensemble strategies [25,26,27,28], direct application of CNN architectures [29,30,31], and the fusion of transfer learning with innovative model designs [32,33,34]. Inspired by these pioneering works, our study aimed to build upon existing methodologies and enhance performance on the aforementioned datasets through innovative techniques and meticulous experimentation. This research addresses histopathology image analysis limitations through advanced deep learning algorithms, focusing on breast cancer classification using VGG16 and ResNet50 architectures. These CNN models were selected for their ability to extract distinctive features crucial for accurate classification. Employing an image patching technique, each image was divided into smaller patches for focused analysis, thus managing computational demands. Initial experiments achieved a promising 89.84% accuracy in patch-level classification. This study provides a comprehensive investigation into deep learning for breast cancer histopathology classification, detailing the methodology, including image patching and CNN implementation, alongside ensemble learning techniques. The results, analysis, and discussions highlight the approach’s efficacy in surpassing the performance of existing studies on the two specified datasets and potential implications for breast cancer diagnosis improvement.

## 2. Materials and Methods

### 2.1. Dataset and Implementation Details

The BACH dataset utilized in this study comprises hematoxylin and eosin (H&E)-stained breast histology microscopy images and whole-slide images. Hematoxylin and eosin (H&E)-stained breast histology microscopy images were valuable in this study because they provided detailed visual information regarding the cellular structure, morphology, and spatial distribution of tissues within the breast, aiding in the interpretation of pathological features relevant to breast cancer research.

In the BACH dataset, we focused on the microscopy images, which are meticulously labeled by medical experts based on the predominant cancer type present in each image. Specifically, images are categorized into four classes: Normal, Benign, In Situ Carcinoma, and Invasive Carcinoma (see Figure 1). The annotation process involved the assessment of each image by two medical experts, with any disagreements resulting in the exclusion of the respective image from the dataset. In total, the microscopy image dataset consisted of 400 images, evenly distributed across the four classes, with each class containing 100 images. These images were stored in .tiff format and possessed specific characteristics: a color model of red–green–blue (RGB), dimensions of 2048 × 1536 pixels, and a pixel scale of 0.42 µm × 0.42 µm. Additionally, the memory space required for each image ranged from 10 to 20 MB, approximately. Notably, the labeling of the dataset was image-wise, providing comprehensive information about the cancer type represented in each image. This meticulously curated dataset served as a valuable resource for training and evaluating machine learning models for breast cancer histopathology image analysis.

The BreakHis dataset consists of 9109 microscopic images of breast tumor tissue. These images were collected from 82 patients and captured at various magnification factors (40×, 100×, 200×, and 400×). The dataset includes 2480 benign and 5429 malignant samples. Each sample is a 700 × 460 pixel, 3-channel RGB image with an 8-bit depth in each channel, stored in PNG format. The dataset categorizes tumors into benign and malignant classes, each with several subclasses. For benign tumors, these subclasses include Adenosis, Fibroadenoma, Tubular adenoma, and Phyllodes adenoma. For malignant tumors, the subclasses include Ductal carcinoma, Lobular carcinoma, Mucinous carcinoma, and Papillary carcinoma. Figure 2 displays eight images, with the first column representing four types of malignant tissues and the second column representing four types of benign tissues.

### 2.2. Data Preprocessing

Breast cancer histopathology images, often characterized by large dimensions and high-resolution details, present challenges in computational analysis. To address this, an image patching technique was used on the BACH dataset to segment the images into smaller, more manageable patches. Each original image, with dimensions of 2048 × 1536 pixels and a pixel scale of 0.42 µm × 0.42 µm, underwent partitioning into six smaller images. This approach effectively reduced the computational burden and memory requirements, which typically ranged from 10 to 20 MB per image. Moreover, employing image-wise labeling facilitated the categorization of patches into distinct classes, including Benign, In Situ, Invasive, and Normal. The patching algorithm involved dividing the image into two halves horizontally and further segmenting each half into three parts vertically. Subsequently, each patch was resized to a target size of 256 × 256 pixels and saved individually. This patching strategy not only facilitated efficient processing but also allowed for focused analysis on specific regions of interest within the histopathology images, thereby enhancing classification accuracy.

In the BreakHis dataset, no preprocessing was applied. The data, which included four scales (40×, 100×, 200×, and 400×), was used to train a single model using multiscale magnification. The data were divided into five folds, with each fold containing all four scale magnifications. There were around 39,545 images of all magnifications, each with a size of 3 × 700 × 460, divided into five folds.

### 2.3. Implementation of VGG16 and ResNet50 Architectures

For our image classification tasks, two powerful convolutional neural network (CNN) models, VGG16 and ResNet50, were utilized. These models have been proven effective in various computer vision tasks and provided a solid foundation for our classification efforts.

The VGG16 architecture is a well-known CNN model comprising multiple convolutional and max-pooling layers, followed by fully connected layers. To expedite our implementation, we employed transfer learning by utilizing pre-trained weights from the ImageNet dataset. We initialized the VGG16 model without its top classification layers and froze the weights of the convolutional layers to preserve the learned features. Custom fully connected layers were then added on top of the VGG16 architecture for classification purposes. During training, we fine-tuned these custom layers while keeping the convolutional layers frozen. This approach facilitated efficient training on our breast cancer histopathology dataset, resulting in accurate classification results.

In addition to VGG16, we incorporated the ResNet50 architecture, another prominent CNN model, into our classification framework. ResNet50 is characterized by its deep structure with skip connections, which helped alleviate the vanishing gradient problem and enabled the training of very deep neural networks. Similar to our approach with VGG16, we loaded the pre-trained ResNet50 model without its classification layers and froze the weights of the convolutional layers to retain learned features. Custom fully connected layers were then added for classification purposes. By leveraging ResNet50, we aimed to capture intricate features from the breast cancer histopathology images, potentially enhancing classification performance compared to shallower architectures.

### 2.4. Implementation of Ensemble Learning

To enhance predictive performance, an ensemble learning approach was adopted by combining predictions from both the VGG16 and ResNet50 models. The ensemble model leveraged the predicted probabilities generated by the individual models. The implementation involved loading the pre-trained VGG16 and ResNet50 models and creating a new model that combined their outputs using averaging. The ensemble model was then compiled with appropriate loss and optimization functions. By harnessing the complementary strengths of VGG16 and ResNet50, our ensemble model aimed to achieve enhanced classification accuracy and robustness, contributing to more reliable breast cancer histopathology image analysis.

Figure 3 presents a schematic of the proposed ensemble model. The model configurations for ResNet34 and VGG16 were as follows: both models were pre-trained and designed for input image size (256 × 256 × 3) to obtain *N* classes. Specifically, in ResNet34, the last fully connected layer was modified to obtain *N* features from the 512 input features (ResNet34 pre-trained model features), whereas the last classifier layer in VGG 16 was modified to obtain *N* features from the 1000 input features (VGG16 pre-trained model features). Both models utilized the CrossEntropyLoss function and the SGD optimizer with a learning rate of 0.001 and momentum of 0.9.

## 3. Results

In this section, the classification outcomes obtained from our study on two distinct datasets are presented. The performance of machine learning algorithms, including VGG16, ResNet, and EfficientNet, is evaluated in comparison with our proposed novel model in this research.

On the BACH dataset, three architectures, VGG16, ResNet, and an ensemble model, were trained and evaluated. Their performance was assessed using various metrics, including training and validation accuracies, obtained through both single-model and ensemble approaches. Table 1 provides a comprehensive summary of the training and validation accuracies, as well as the losses for 5-fold cross-validation, achieved by each architecture on the BACH dataset.

The VGG16 architecture demonstrated commendable results, achieving a training accuracy of 98.72% and a validation accuracy of 79.95%, as depicted in Figure 4a. Similarly, the ResNet architecture exhibited strong performance, with a training accuracy of 99.05% and a validation accuracy of 80.47%, as illustrated in Figure 4b. Notably, the ensemble model, trained with 5-fold cross-validation, showcased superior performance, boasting a training accuracy of 99.84% and a validation accuracy of 92.58%, as indicated in Figure 4c. Furthermore, the ensemble model trained without cross-validation attained the highest accuracy, with a training accuracy of 99.78% and a validation accuracy of 95.31%, as illustrated in Figure 4d.

The notable improvement observed with the ensemble model compared to individually modified CNN architectures can be attributed to its ability to combine the strengths of individual architectures while mitigating their weaknesses. By leveraging the diversity among models generated through different initialization and training strategies, the ensemble model achieved robust predictions. Additionally, the incorporation of cross-validation helped generalize the model, reducing overfitting and enhancing its performance on unseen data. These results underscore the efficacy of ensemble learning in enhancing classification accuracy, particularly in the context of medical image analysis, where subtle variations in image features can significantly impact diagnosis. Several studies have implemented machine learning algorithms for the BACH dataset for breast cancer histology image classification, achieving varying levels of accuracy. For instance, the top submissions in the BACH 2018 Grand Challenge used convolutional neural networks (CNNs) and reported an accuracy of 87% [8]. Another study by Song et al. (2020) applied a DenseNet model combined with data augmentation techniques and reported an accuracy of 88.7% [35]. Similarly, Araujo et al. (2019) employed a CNN-based approach, achieving an accuracy of 85.6% [36]. These studies highlight the effectiveness of deep learning methods in improving the accuracy of breast cancer histology classification using the BACH dataset.

On the BreakHis dataset, the ensemble model demonstrated superior performance compared to state-of-the-art algorithms in terms of accuracy and Jaccard index for five-fold cross-validation. Table 2 presents a comparative analysis of various machine learning models, including EfficientNet b0, VGGNet 16, ResNet 34, ResNet 50, and an ensemble model. These models were evaluated based on their training and validation performance metrics such as accuracy, Jaccard index, and loss. Remarkably, the ensemble model outperformed all other models, with the highest average validation accuracy of 98.43%, indicating its superior predictive capability on unseen data. Various studies have implemented machine learning algorithms for the BreaKHis dataset, showcasing a range of accuracies. Simonyan, K et al. proposed a three-path ensemble architecture combining VGG19, MobileNetV2, and DenseNet201, which resulted in a 97% classification accuracy [14]. Additionally, Howard et al. proposed a stacked ensemble learning framework and achieved 94.35% accuracy, while a traditional SVM-based method reported an 81.2% accuracy [37]. Also, the recent work presented in İrem Sayın et al.’s comparison study [38] achieved an accuracy of 89% using the Xception model. Both the Inception and InceptionResNet models achieved an accuracy of 87%. These results demonstrate the effectiveness of different machine learning models on the BreaKHis dataset, highlighting significant advancements in breast cancer classification accuracy.

Table 3 presents an analysis of the Ensemble model designed for the classification of breast histopathological images from the BreakHis dataset into benign and malignant categories. This model achieved an exceptional average validation accuracy of 99.72%, significantly surpassing the performance of all prior models referenced in Table 4. This can be seen with BreastNet [18], which achieved an overall accuracy of 97.56%. Similarly, DenseNet121-AnoGAN [19], an unsupervised anomaly detection model using generative adversarial networks, recorded an accuracy of 91.44%. Transfer learning applications on the BreakHis dataset with ResNet-18 [20] and Inception-V3 [21] demonstrated accuracies of 92.15% and 98.97%, respectively. Additionally, ref. [22] proposed an enhanced autoencoder network utilizing a Siamese framework, achieving an accuracy of 96.97%. DRDA-Net [23], a model featuring dual-shuffle attention mechanisms, attained an accuracy of 96.10%, and DenTnet obtained an accuracy of 99.28%. Figure 5 shows the ROC curve of the ensemble model for binary classification on the BreakHis dataset. The ROC curve shows an excellent performance of the proposed model for classifying benign and malignant blocks.

The success of the ensemble model on the BreakHis dataset can be attributed to its capacity to capture intricate patterns and features present in histopathology images, which are crucial for accurate diagnosis. Furthermore, the ensemble approach helps alleviate the challenges posed by dataset heterogeneity and variability, leading to robust and reliable predictions across diverse tissue samples. These findings highlight the potential of ensemble learning to address the complexities inherent in medical image classification tasks, thereby facilitating more accurate and timely diagnoses.

Overall, our results demonstrate the effectiveness of deep learning techniques, particularly ensemble learning, in accurately classifying breast cancer histopathology images. These findings underscore the potential of artificial intelligence to enhance breast cancer diagnosis and management, paving the way for improved patient outcomes and reduced healthcare burdens. The training and validation were conducted using an Nvidia Tesla V100 graphics card. The ensemble learning architecture was trained over 20 epochs, with each epoch comprising 48 steps. On average, each step of training took approximately 16 s. Additionally, our study highlights the importance of exploring novel approaches and ensemble strategies in medical image analysis, offering promising avenues for future research and clinical applications.

## 4. Discussion

The results obtained from our comprehensive study on breast cancer histopathology image classification underscore the pivotal role of advanced computational techniques, particularly deep learning algorithms, in improving diagnostic precision and efficiency. Various aspects, including computational efficiency, algorithmic advancements, and future directions for leveraging these methodologies in clinical practice and beyond, were discussed.

### 4.1. Computational Efficiency

One of the key challenges in histopathology image analysis lies in handling large-scale datasets with high-resolution images, which often pose computational burdens. To address this challenge, innovative techniques such as image patching, downsampling, and ensemble learning were employed. The approach of segmenting high-resolution images into smaller patches enabled focused analysis of localized regions, reducing computational demands while preserving diagnostic accuracy. Additionally, pre-trained convolutional neural network (CNN) models like VGG16 and ResNet50 were leveraged to harness learned features, thereby optimizing computational efficiency. Furthermore, the ensemble learning strategy, which combined predictions from multiple models, not only enhanced classification accuracy but also mitigated computational costs by leveraging existing model architectures.

### 4.2. Algorithmic Advancements

This study contributes to the ongoing advancements in algorithmic design for breast cancer histopathology image classification. By exploring diverse CNN architectures and ensemble learning techniques, the efficacy of leveraging deep learning for accurate classification across multiple datasets was demonstrated. The utilization of pre-trained models, fine-tuned strategies, and ensemble approaches enabled the extraction of intricate features from histopathology images, leading to enhanced classification performance. Moreover, the investigation into multiscale analysis, particularly for the BreakHis dataset, highlighted the importance of integrating information from different magnification levels to capture diverse pathological structures and improve diagnostic accuracy. This approach enhanced the performance of image classification by integrating information at different magnifications, thereby improving the recognition of various pathological structures that are more discernible at certain scales. It also increased the efficiency and accuracy of diagnosis by reducing operator variability. Moreover, multiscale analysis can assist in understanding disease progression for the development of targeted therapeutics. Automated systems that employ this technique can reduce the time required for image analysis. Consequently, multiscale magnification and can offer a comprehensive view of histopathological images and facilitate a more precise and efficient disease diagnosis. These algorithmic advancements also hold promise for accelerating the research and development of personalized treatment strategies.

### 4.3. Future Directions

Looking ahead, several avenues warrant exploration to further enhance the utility of deep learning in breast cancer diagnosis and management. Firstly, the development of user-friendly web applications incorporating our classification algorithms could empower healthcare professionals with accessible tools for rapid and accurate histopathology image analysis. Promising studies that validate the efficacy of such AI-aided tools would be preferred to create viable applied solutions to enhance healthcare workflows [41,42]. Such applications could streamline clinical workflows, facilitate remote consultations, and improve patient care outcomes. Additionally, extending our algorithmic framework to encompass other types of cancer, such as lung cancer or prostate cancer, may hold promise for broadening the scope of AI-driven diagnostic solutions. Additional imaging modalities could also be leveraged to encompass end-to-end care paths aided by AI solutions, as explained by the authors of a review study [43]. By leveraging transfer learning and dataset adaptation techniques, our methodologies could be adapted to address diverse oncological challenges, thereby advancing cancer research and clinical practice. Furthermore, integrating multimodal data, such as genomic profiles or clinical metadata, into our classification models could enrich diagnostic insights and enable more comprehensive patient stratification.

## 5. Conclusions

In conclusion, our research represents a significant stride towards leveraging advanced computational techniques, particularly deep learning algorithms, to enhance breast cancer diagnosis and management through histopathology image analysis. By focusing on two prominent datasets, BACH and BreakHis, and employing innovative methodologies, a substantial contribution is made to the growing body of knowledge aimed at improving diagnostic precision and efficiency in breast cancer detection. Through the meticulous experimentation and implementation of deep learning models, including VGG16 and ResNet50, the efficacy of this approach in accurately classifying breast cancer histopathology images was demonstrated. The proposed ensemble learning strategy, which combines predictions from multiple models, showcased superior performance compared to state-of-the-art algorithms, underscoring the importance of leveraging the complementary strengths of different CNN architectures.

Furthermore, our research addressed the computational challenges inherent in histopathology image analysis by introducing techniques such as image patching and multiscale analysis. These strategies enhance computational efficiency and facilitate focused analysis on specific regions of interest within the histopathology images, improving classification accuracy.

## Figures and Tables

**Figure 1 cancers-16-02222-f001:**
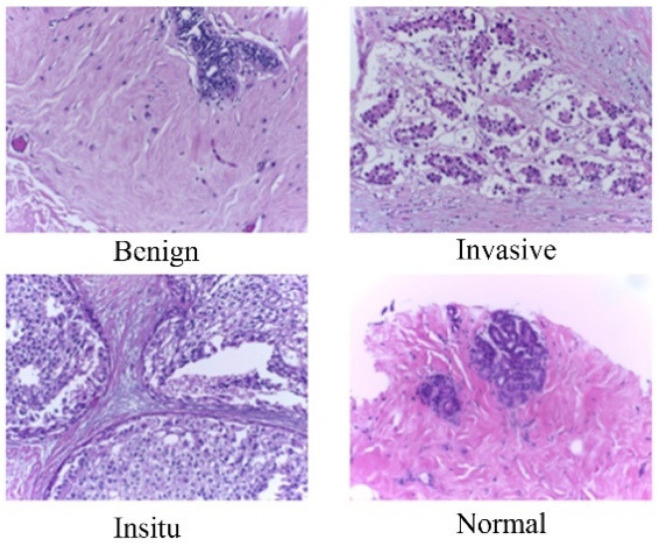
Exemplary microscopy images demonstrating the four classes in the BACH dataset (on 40× magnification).

**Figure 2 cancers-16-02222-f002:**
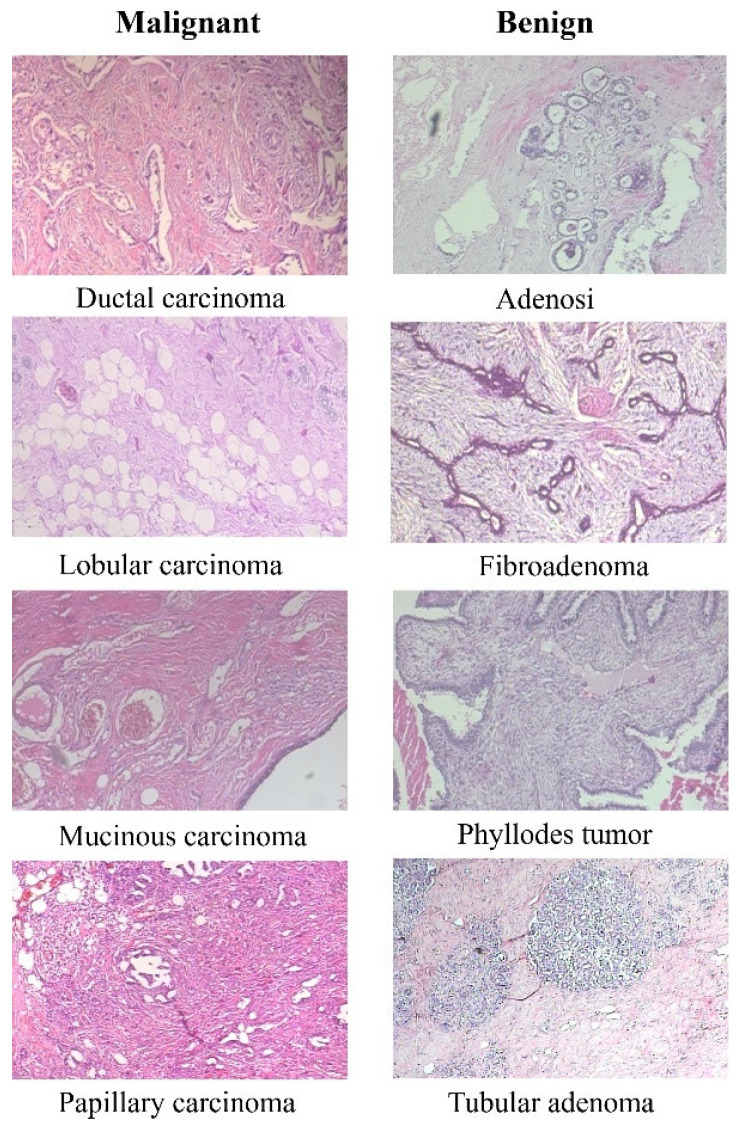
Representative microscopy images of malignant and benign breast tissues from the BreakHis dataset (on 40× magnification).

**Figure 3 cancers-16-02222-f003:**
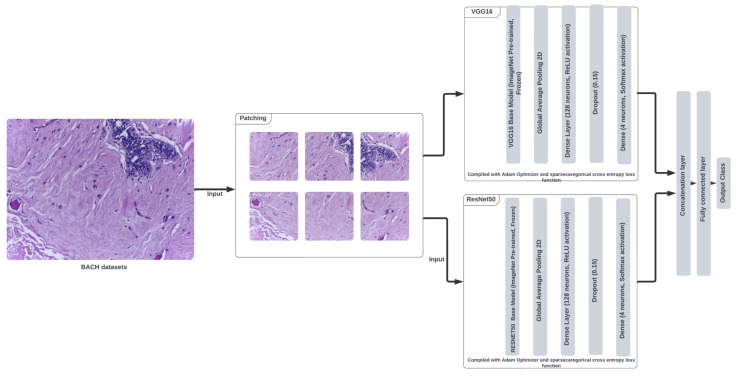
Schematic of ensemble model.

**Figure 4 cancers-16-02222-f004:**
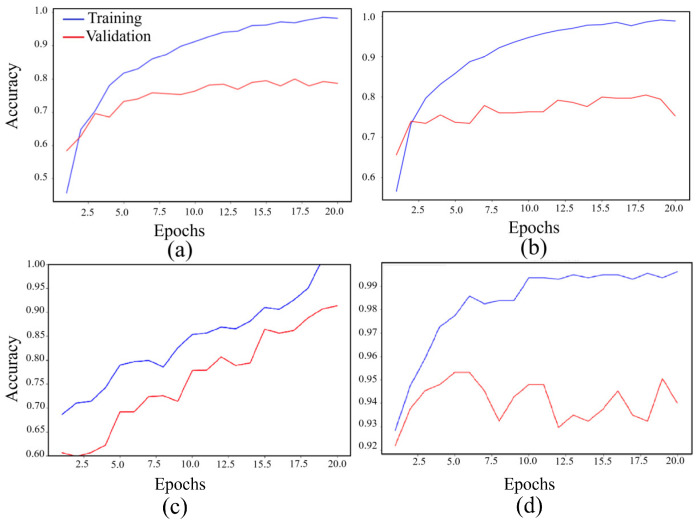
The average accuracy on 5-fold cross-validation on BACH dataset for the following architectures: (**a**) VGG 16 model, (**b**) ResNet 50 model, (**c**) ResNet 34 model, and (**d**) ensemble model.

**Figure 5 cancers-16-02222-f005:**
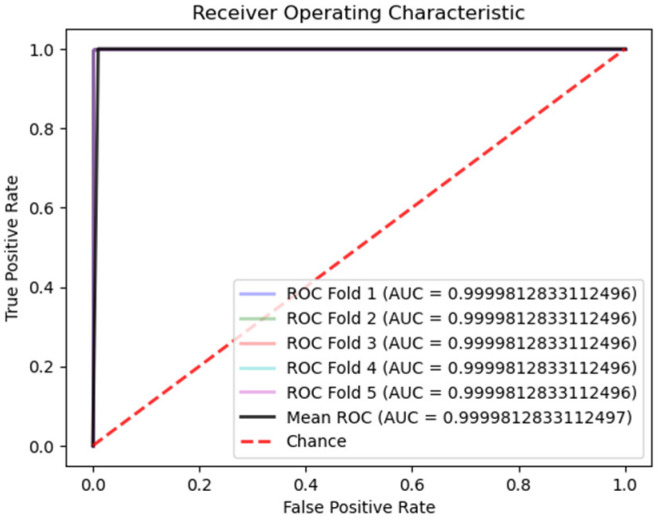
ROC curve of the ensemble model for five-fold cross-validation.

**Table 1 cancers-16-02222-t001:** Comparative performance analysis of machine learning models for breast cancer diagnosis on BACH dataset.

Model	Average Train Accuracy	Average Validation Accuracy	Train Loss	Average Validation Loss
VGGNet 16	98.72	79.95	0.0966	0.7399
ResNet 50	99.05	80.47	0.0564	0.4489
Ensemble Model (5-fold cross-validation)	99.84	92.58	0.0591	0.2793

**Table 2 cancers-16-02222-t002:** Comparative performance analysis of machine learning models for breast cancer diagnosis on BreakHis dataset.

Model	Train Accuracy	Train Jaccard Index	Val Accuracy	Val Jaccard Index	Average Accuracy
EfficientNet B0	79.69402	0.673097	84.95638	0.75735	84.95
VGGNet 16	96.91175	0.937524	97.38273	0.955309	97.38
ResNet 34	97.75888	0.962849	98.22481	0.96945	98.22
ResNet 50	97.4744	0.957038	97.30434	0.953921	97.30
Ensemble Models (ours)	98.41952	0.973193	98.42964	0.973642	98.43

**Table 3 cancers-16-02222-t003:** Binary classification performance of ensemble models on BreakHis dataset.

Model	Train Accuracy	Train Jaccard Index	Val Accuracy	Val Jaccard Index	Average Accuracy
Ensemble Models (ours)	99.76887	0.995616	99.71931	0.994532	99.71930712

**Table 4 cancers-16-02222-t004:** Binary classification performance on BreakHis dataset.

Year	Method	Accuracy
2020	Togacar et al. [18]	97.56
	Parvin et al. [39]	91.25
	Man et al. [19]	91.44
2021	Boumaraf et al. [20]	92.15
	Soumik et al. [21]	98.97
2022	Liu et al. [22]	96.97
	Zerouaoui and Idri [40]	93.85
	Chattopadhyay et al. [23]	96.10
	DenTnet [17]	99.28
2024	Ensemble Models (ours)	99.72

## Data Availability

The data presented in this study are available in this article.

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
