# Peer review of "Ensemble Deep Learning-Based Image Classification for Breast Cancer Subtype and Invasiveness Diagnosis from Whole Slide Image Histopathology"

_cancers, 2024, doi:10.3390/cancers16122222_

Round 1

Reviewer 1 Report

Comments and Suggestions for Authors

For BC subtyping, the authors employes VGG16 and ResNet50 models in an ensemble strategy, enhanced by a novel image patching technique for high-resolution images for the BACH dataset. For the BreakHis dataset, the model uses VGG16, ResNet34, and ResNet50 models to classify images into eight categories, split evenly between benign and malignant. A five-fold cross-validation method was applied to both datasets for thorough training and testing. The results show a patch classification accuracy of 95.31% for BACH and a WSI classification accuracy of 98.43% for BreakHis. The work has some potentials, however, I have the following concerns:

-Most of the folds show an overfitting at/after epoch 18, how they stop the model.

-More information about the hyper-parameters are required.

-Table 4, that is the point of mentioning the non-reporting results works?

- AUROC curve is required to show the performance of the work on multiple running points.

Author Response

Dear reviewer,

I hope this email finds you well. I am writing to express my sincere gratitude for the time and effort you dedicated to reviewing my research article titled “Ensemble Deep Learning-Based Image Classification for Breast Cancer Subtype and Invasiveness Diagnosis from Whole Slide Image Histopathology.” Your insightful comments and constructive feedback have significantly contributed to improving the quality of the manuscript.

Your expertise and knowledge in the field have been invaluable in refining this research. Your thoughtful suggestions have helped shape the paper into its best possible form. I truly appreciate your attention to detail and the thoroughness with which you reviewed our work.

Once again, thank you for your invaluable contribution to this research. Your feedback has been instrumental in enhancing the scientific rigor and impact of our study.

Best regards,

Dr. Salah Alheejawi.

Northeastern University, College of Engineering, United States; [email protected]

Reviewer 2 Report

Comments and Suggestions for Authors

The manuscript “Ensemble Deep Learning-Based Image Classification for Breast Cancer Subtype and Invasiveness Diagnosis from Whole Slide Image Histopathology” revealed a comprehensive approach utilizing ensemble deep learning techniques to analyze breast cancer histopathology images. The manuscript is well documented and explains the developed assay very well. However, some areas of the manuscript need to be improved comprehensively. The suggested comments need to be addressed before I can recommend this work for publication.

*It is better to draw schematic illustrations of the manuscript for better understanding.

*Tables need justification (Format) and enhance the resolution of figures.

*Need to compare with the previous method.

*The overall language needs improvement, as there are numerous ambiguities. It is better to use passive language for discussion.  

*The whole introduction needs to be modified to make linkage for a better understanding of previous studies, gaps and the need of the present study.

* Check the reference, references need consistency with format.

Comments on the Quality of English Language

*The overall language needs improvement, as there are numerous ambiguities. It is better to use passive language for discussion. 

Author Response

(The authors gave the same response as above.)

Round 2

Reviewer 1 Report

Comments and Suggestions for Authors

The manuscript is in a very good shape now.